# Differences between Squamous Cell Carcinomas of the Base of the Tongue and the Tonsils in Prevalence of HPV16 Infection, Its Type, and Clinical Features

**DOI:** 10.3390/jpm13020361

**Published:** 2023-02-18

**Authors:** Beata Biesaga, Anna Mucha-Małecka, Anna Janecka-Widla, Krzysztof Małecki

**Affiliations:** 1Department of Tumor Pathology, Maria Skłodowska-Curie National Research Institute of Oncology, Cracow Branch, Garncarska 11, 31-115 Cracow, Poland; 2Department of Radiotherapy, Maria Skłodowska-Curie National Research Institute of Oncology, Cracow Branch, Garncarska 11, 31-115 Cracow, Poland; 3Department of Radiotherapy for Children and Adults, University Children’s Hospital of Cracow, Wielicka 265, 30-663 Cracow, Poland

**Keywords:** cancers of base on tongue, cancers of tonsil, difference in HPV infection, difference in clinical features

## Abstract

Regarding attempts to find de-escalation methods of treatment for patients with HPV16-positive squamous cell carcinoma of the oropharynx (OPSCC), there is an urgent need to identify new prognostic factors which allow physicians to differentiate the prognosis of these patients. The aim of the study is to compare the incidence of transcriptionally active HPV16 infection and its type as well as other epidemiological, clinical, and histopathological features between SCC of the base of the tongue (BOTSCC) and tonsils (TSSCC). The analysis was performed in a group of 63 patients with OPSCC, for which, in our earlier studies, we assessed transcriptionally active HPV16 infection and its type (viral load and viral genome status). Transcriptionally active HPV16 infection was significantly more common in TSSCC (96.3%) than in BOTSCC (3.7%). Patients with TSSCC had significantly higher disease-free survival rates (84.1%) than those with BTSCC (47.4%); the same was true in the subgroup with HPV16 positivity. The obtained results are an important indication for further research on the development of new prognostic and/or predictive factors for patients with HPV16-positive squamous cell carcinomas of the oropharynx.

## 1. Introduction

Head and neck tumors (HN) are located in the upper gastrointestinal and respiratory tract, mostly in the oral cavity, oropharynx, and larynx. Around 95% of these tumors are squamous cell carcinomas (SCC). Heavy smoking, betel chewing, alcohol consumption, and human papillomavirus (HPV) infection are listed as the main etiological factors for these tumors. HPV infection (mainly HPV16) applies particularly in the case of squamous cell carcinomas of the oropharynx (OPSCC), which includes cancers localized on the base of the tongue, tonsils, soft palate, and pharyngeal wall. Over the last several years, an increase in incidence rates for SCC of the tonsil (TSSCC) and base of tongue (BOTSCC) has been observed, which is probably related to the increased number of HPV infections in these anatomical sites [1].

Numerous clinical studies have revealed that patients with HPV16-dependent OPSCC have a better prognosis than uninfected patients ([2]—review). These results inspired the impulse to undertake clinical trials into the possibility of de-escalation of anti-cancer treatment in patients with HPV16-dependent OPSCC in order to avoid toxic reactions after cisplatin or/and radiotherapy treatment. In these studies, various de-escalation strategies were or are still being tested, such as a decrease of the dose of radiotherapy (RT), reduction of the irradiation area, replacement of cisplatin (CisPt) during concurrent chemoradiotherapy (CRT) with less toxic systemic agents (e.g., cetuximab), or selection of patients for concurrent adjuvant chemoradiotherapy based on the results of induction chemotherapy ([3]—review). However, in two completed phase III clinical trials (RTOG 1016 and De-ESCALaTE) concerning randomly assigned patients with HPV16-dependent OPSCC for radiotherapy with concurrent cetuximab or CisPt, a significantly better OS) and higher local cure rate were observed in the CisPt arm [4,5]. In light of these results, it should be noticed that in more than 40% of patients with HPV16 positive OPSCC, progression of cancer disease is noted [6,7,8,9]. In addition, in the aforementioned clinical trials on de-escalation of treatment, de-intensification schemes were applied for all patients with viral infections. Therefore, further improvement of treatment outcomes may be related to the possibility of differentiating the prognosis of patients with HPV16-dependent OPSCC and, thus, the need to develop new prognostic and/or predictive factors. These factors would allow physicians to distinguish patients with viral infections that have a better prognosis (for whom treatment de-escalation should be applied) and those who have a worse prognosis (for which de-escalation of treatment is not possible).

Currently, there are attempts to select patients with OPSCC with HPV positivity for de-escalation treatment. Both clinical and biological features are taken into account. With regard to clinical features, the following ones are considered to be risk factors: clinical advancement stages T4 and N3, bilateral lymph node involvement, high level of smoking, and no response to induction chemotherapy [10,11]. In turn, few preclinical studies have shown that the presence of mutations in *phosphatidylinositol 3-kinase* gene [12], overexpression of the phosphorylated form of AKT protein kinase (Thr308) [12], lack of estrogen overexpression [13] and high level of mutant-allele tumor heterogeneity [14,15] are associated with a worse prognosis of OPSCC patients with HPV16 infection. In our earlier report on the subgroup of patients with HPV16-positive OPSCC, we also found a lack of CD44 overexpression and a lack of Sox-2 overexpression as independent positive prognostic factors, whereas in the HPV16 negative subgroup, these features had no prognostic potential [16]. In turn, the meta-analysis of Ndiaye et al. [17] has shown a slightly higher prevalence of HPV infection in TSSCC (53.9%) compared to BOTSCC (47.8%). A recent report also suggests that there are differences in epidemiological and clinical features (age, gender, race, facility, TN stages) between BOTSCC and TSSCC [18]. In the literature on the subject, to the best of our knowledge, there are no studies on the differences in the occurrence of a specific type of HPV16 infection (distinguished by viral genome state: integrated vs. episomal vs. mixed and the viral load (VL—the number of viral copies per cell)) between BOTSCC and TSSCC, while their demonstration may contribute to the optimization of treatment for patients with HPV16-positive OPSCC.

In our earlier studies on the group of 63 patients, we assessed transcriptionally active HPV16 infection (based on analysis of P16 expression and HPV16 DNA presence) [12] and its type (viral genome state and VL) [19]. In this group, active HPV 16 infection was noticed in 27 OPSCC (42.9%). It should be noted that detection of HPV infection based only on the analysis of virus DNA or P16 expression may lead to obtaining false positive results. HPV DNA presence does not indicate whether HPV is transcriptionally active or not. P16, in turn, is a surrogate marker indicating active HPV infection, but its overexpression may not exactly match the HPV DNA because it may also be caused by other non-viral factors. Among the 63 tumors, those with mixed states of genomes (70.4%) and higher VL (63.0% > 6764.3 viral copies/cell) prevailed. Thus, the main aim of the present study is to compare the frequency of transcriptionally active HPV16 infection and its type as well as epidemiological, clinical, and histopathological features between BOTSCC and TSSCC in the group of 63 patients with OPSCC. 

## 2. Materials and Methods

### 2.1. Study Design

This study was performed on a group of 63 patients with squamous cell carcinoma of the oropharynx included in the analysis according to the following criteria: (1) SCC of the oropharynx, (2) lack of previous cancer treatment, (3) no distant metastases at the time of diagnosis and (4) follow-up time no shorter than 5 years until the end of 2019. The research was conducted based on formalin-fixed and paraffin-embedded tumor tissue fragments (FFPE) obtained for each patient prior to the initiation of cancer treatment. Before cutting for DNA isolation and immunohistochemistry, all blocks have undergone histopathological verification based on eosin/hematoxylin-stained slides to confirm the histological diagnosis, grade, and degree of keratinization. For further analysis, histopathologists indicated paraffin blocks with at least 50% of tumor neoplasm. All other details concerning inclusion and exclusion criteria as well as histopathological verification, were presented earlier [12].

### 2.2. DNA Extraction

DNA was extracted from 4 µm thick 3–5 sections using ReliaPrep FFPE gDNA Miniprep System (Promega, WI, USA) according to manufacturer suggestions with our own modification concerning prolongation of digestion at 56 °C from 1 h to overnight to provide improved quantity and quality of DNA. Deparaffinization was performed using mineral oil at 80 °C. After adding lysis buffer and centrifugation, the sample was digested with Proteinase K. Samples were further incubated for 1 h at 80 °C, digested with RNase A, and mixed with BL buffer and 96% ethanol. After centrifugation, the entire aqueous phase containing DNA was transferred into the binding column, washed twice, and finally eluted. DNA concentration and purity (measured as A260/280 and A260/230 ratios) were evaluated spectrophotometrically with Biophotometer Plus (Eppendorf, Germany). Samples were stored at −20 °C until used. A more detailed description of this procedure was presented earlier [20].

### 2.3. Assessment of Transcriptionally Active HPV16 Infection

To screen OPSCC samples for the presence of HPV DNA, nested PCR was applied, which consisted of two successive PCR runs, and the product of the first reaction with PGMY09/PGMY11 serves as a template in the second run with GP5+/GP6+. PGMY09/PGMY11 primers allow the amplification of the *L1* gene fragment of multiple HPV types simultaneously, however, without indication of virus type precisely. A full list of primer sequences as well as all details concerning the composition of the reaction mixture or parameters of PCR runs, were reported earlier [12]. Negative (water) and positive (DNA isolated from HPV-positive cervical cancer tissue) controls were added to each run. The final products were separated in 2% agarose gel and visualized under UV light using SimplySafe dye (EURx, Poland). For each tumor, three independent analyses were performed, and a sample was classified as HPV DNA positive when at least one positive signal was observed.

HPV genotyping was performed for samples classified as HPV DNA positive in nested PCR. Genotyping was performed using AmoyDx Human Papillomavirus (HPV) Genotyping Detection Kit (Amoy Diagnostics Co., Xiamen, China) and quantitative PCR (qPCR). This assay identifies 19 high-risk HPV DNA (16, 18, 26, 31, 33, 35, 39, 45, 51, 52, 53, 56, 58, 59, 66, 68, 70, 73, and 82) and two low-risk HPV DNA (6 and 11). The procedure was performed according to the suggestions of the manufacturer. More details about this procedure are described by us earlier [12].

P16 immunostaining was performed in the subgroup of HPV16-positive tumors using CINtec^®^ Histology Kit (Roche, Heidelberg, Germany) according to the manufacturer’s procedure described by us earlier [19]. In brief, 4 µm thick sections of FFPE HNSCC tissues were deparaffinized and hydrated through a series of xylenes and alcohols. After antigen unmasking (96 °C, 10 min) and exogenous peroxidases quenching (5 min), sections were incubated with primary anti-P16 antibody (clone E6H4, RT, 30 min) followed by 30 min incubation with visualization system. P16 was visualized using 3,3′–diaminobenzidine, and for nuclear counterstaining, hematoxylin was applied. Cervical cancer tissue with P16 overexpression was used as a positive control. For negative control, the primary antibody was omitted. Immunopositivity was defined according to Lewis et al. [21] as follows: >75% of positive staining cells or >50% staining with >25% confluent areas of positive staining.

Tumors were defined as those with transcriptionally active HPV16 infection if they contained HPV DNA (detected during nested PCR and then confirmed by genotyping assay) and overexpressed P16 (according to immunohistochemistry). All other cases (HPV DNA+/P16−, HPV DNA−/P16+, and HPV DNA−/P16−) were classified as those lacking a transcriptionally active HPV16 infection [12].

### 2.4. Assessment of Viral Load and Viral Genome Status

HPV16 VL and the type of its genome state were determined by qPCR. For VL analysis, two standard curves were obtained. One of them concerns Ct for a 139-bp fragment of *β*-*actin* gene in serial 10-fold dilutions of the reference human genomic DNA (Roche Diagnostics, GmbH, Germany) vs. the number of gene copies to obtain the number of cells in the sample. The second one concerns Ct for HPV16 *E6* in serial 10-fold dilutions of the HPV16 plasmid (ATCC, USA) vs. the number of gene copies to obtain the number of HPV16 copies in the sample. HPV16 VL was calculated as the number of virus copies per cell, assuming that two copies of the *β-actin* gene correspond to one cell. All other details concerning these procedures were presented earlier [19].

Physical HPV16 genome state (episomal, integrated, mixed) was analyzed on the basis of the Ct*E2*/Ct*E6* ratio. The viral genome was regarded as integrated when the Ct*E2*/Ct*E6* was 0, as episomal when this ratio was 1 or more, and as mixed when the Ct*E2*/Ct*E6* was between 0 and 1 as we described earlier [22].

### 2.5. Statistical Analysis

Descriptive statistics were used to determine the mean and median values of continuous variables and standard errors of means (SE). Associations between categorical variables were analyzed using the Pearson Chi-square test. In survival analysis, as an endpoint, disease-free survival was applied as a time from the end of therapy until the first documented evidence of recurrent disease—treatment failure, locoregional recurrence, distant metastasis, within 5 years after completing the treatment. In this analysis, differences between subgroups were tested by the log-rank test. All statistical tests were two-sided, and *p* < 0.050 was considered significant. Statistical analyses were carried out using the Statistica program, version 13.0.

## 3. Results

### 3.1. Clinical Characteristics of a Group of 63 Patients with Squamous Cell Carcinoma of the Oropharynx

The subgroup of 63 patients with OPSCC included 15 women (23.8%) and 48 men (76.2%) aged from 38 to 75 years, with a mean of 57.5 years ± 1.2 (SE) (Table 1). In the analyzed group, T3 (50.8%), N2 (55.6%), non-keratinized (55.6%), and G2 tumors (52.4%) predominated. The majority of patients were treated with CRT-CisPt (n = 28, 44.4%), either alone (n = 22, 78.6%) or after surgery (n = 6, 21.4%). In 19 patients (30.2%), radiotherapy was used alone (n = 6, 31.6%) or adjuvantly after surgery (n = 13, 23.3%), while 16 patients (25.4%) were treated with chemotherapy induction (CisPt + 5-fluorouracil + taxanes), followed by ionizing radiation. The decision regarding the operation was made by a team of physicians consisting of a surgeon, radiotherapist, and chemotherapist, who referred for the surgery patients with advanced disease (T ≥ 3) and histological grade G1. A detailed description of the treatment, doses of cisplatin, and RT can be found in our earlier publications [12].

For 63 patients, the mean follow-up time was 42.0 months ± 4.4 and ranged from 0 to 113 months. At the time of the study, 45 patients (71.4%) had regressed, and 18 (28.6%) had progression (2 failures, 12 local recurrences, and 4 distant metastases) 0 to 39 months after completion of the study.

### 3.2. Differences between Squamous Cell Carcinoma of Base of Tongue and Tonsil in Epidemiological, Clinical, and Histopathological Features as Well as Prevalence of HPV16 Infection and Its Type

In the group of 63 patients, tonsil cancers predominated (n = 44, 69.8%) (Table 1). Among patients with TSSCC, female gender, lower clinical stage T (T2 and T3), and lack of keratinization were significantly more frequent as compared to BOTSCC. TSSCC patients were also characterized by a higher rate of cancer regression at 5 years after completing the oncological treatment (80.0%) than patients with BOTSCC (20.0%). There were no other statistically significant differences between SCCTS and SCCBOT in terms of the rest of the analyzed epidemiological, clinical, and histopathological features.

In the group of 63 OPSCC, transcriptionally active HPV16 infection was present in 27 tumors (42.9%), and it was significantly more common in TSSCC (96.3%) than in BOTSCC (3.7%) (Table 1). One patient with BOTSCC and HPV16 infection was a woman, 60 years old, with Karnofsky scale <80%, T3, N2 stages and grade G2, addicted to alcohol and smoking, treated by surgery and adjuvant RT, who developed distant metastases 15 months after completing the treatment. All cancers with a higher VL (median > 6764.3 virus copies per cell) were TSSCC, and all BOTSCC had VL equal to or below the median. There was no significant difference between BOTSCC and TSSCC according to the frequency of specific status of the HPV16 genome (integrated vs. mixed).

In the group of 63 patients with HNSCC, the 5-year DFS was 66.3%. Patients with TSSCC had significantly higher DFS (84.1%) than those with BOTSCC (47.4%) (Table 2, Figure 1). All, except one, TSCC patients with transcriptionally active HPV16 infection survived 5 years without cancer progression (96.1%), whereas, for one patient with BOTSCC and this infection, cancer progression (distant metastasis) was observed at the 15^th^ month after completing the treatment. In the subgroup of HNSCC patients without HPV16 presence, localization of the tumor did not significantly influence DFS.

## 4. Discussion

To the best of our knowledge, this is the first report showing the difference in the level of viral load between the SCC of the base of the tongue and the tonsils.

We have namely shown the significantly increased frequency of cancers with higher VL among TSSCC compared to BOTSCC (Table 1). Moreover, in TSSCC, a significantly higher incidence of transcriptionally active HPV16 infection (Table 1) and better disease-free survival (Table 2) were noticed. This result suggests better DFS for TSSCC patients with transcriptionally active HPV16 infection than for BOTSCC patients with this infection. In fact, in HNSCC patients with HPV16 positivity, tumor localization had a significant impact on DFS (Table 2), although this result should be treated with caution because there was only one patient with BOTSCC and HPV16 infection. However, these results are in line with those obtained by us earlier [19] and other authors [17]. Referring to VL, in our earlier analysis concerning the group of 36 patients with HPV16-positive HNSCC, significantly better OS and DFS for patients with higher VL were shown [19]. As we now know today, the vast majority of these patients suffered from tonsillar cancers. In turn, a higher prevalence of HPV in TSSCC than in BOTSCC is in accordance with the results of many studies, which were summarized in the meta-analysis of Ndiaye et al. [17], in which 148 publications were included. In this meta-analysis, the percentage of HPV prevalence in TSSCC was 53.9%, whereas, in BOTSCC, it was 47.8%. The percentage concerning BOTSCC is much higher than this obtained by us (3.7%). However, we analyzed a much lower number of patients and only transcriptionally active HPV16 infection, whereas the above-mentioned analysis concerns different types of HR-HPV and different methods of HPV DNA detection.

There are three important questions that should be discussed in the context of these results: (1) why HPV infection is noticed more often in the tonsils than in the base of the tongue, (2) why HPV16 infection in the tonsils is characterized by higher VL as compared to BOTSCC and (3) why TSSCC are characterized by better prognosis than BOTSCC. Results of some studies suggest that the answer to all these questions may be related to differences in the functioning of the immune system between BOTSCC and TSSCC. Recently, Chen et al. [23] interrogated the TCGA HNSCC data and distinguished a subgroup of patients referred to as the “active immune class” associated with oropharyngeal tumors, T cell-inflamed signature, and the presence of HPV. Unfortunately, they did not report data concerning the subsite of HNSCC. In turn, Welters et al. [24] detailed the immune compartment in OPSCC with HPV16 positivity. In 64% HPV16-positive tumors, they indicated the presence of type I specific T-cells as well as higher numbers of activated CD161^+^ T cells, CD103^+^, CD103+ tissue-resident T cells, dendritic cells, and DC-like macrophages. In their study, HPV16-positive tumors with specific T-cells had better OS and lower T and N stages. Although they did not differentiate subgroups with BOTSCC and TSSCC, in the light of present results, it can be assumed that patients with cancers characterized by the presence of type I specific T-cells suffer rather from TSSC than from BOTSCC. In regard to higher VL noticed for TSSCC than for BOTSCC in the present study, some reports have found significant correlations between higher VL and overproduction of circulating antibodies against plasma virus-like particles or antibodies against HPV16 oncoproteins E6 and E7 [25] and between higher VL and expression of HPV *E6*/*E7* mRNA [26,27,28]. All these facts may suggest that in TSSCC, transcriptionally active HPV infection and the same higher VL may be related to additional stimulation of immune response. This particular stimulation in TSSCC may indicate an anatomical distinction between TSSCC and BOTSCC. A recent study has shown that the tonsils are comprehensive sources of extrathymic T-cell development and also sites for the development of innate immune cells known as group 3 innate lymphoid cells (ILC3) [29]. These cells are unique to the lamina propria and tonsils and probably constitute an important first line of defense against viral infections. BOT tissue harbors mucus glands which drain directly into crypts rather than the surface. Additionally, crypts are shallower with less complex branching compared with the crypts in palatine tonsils, which may reflect a different immunologic milieu or structure as well [15]. However, contrary to the suggestion about more effective stimulation of the immune system in TSSCC, it was revealed that OPSCC with HPV infection were more likely to be B7-H1 positive (B7-H1 is involved in B7-H1/PD-1 signaling pathway of host immune suppression), what allows to avoid inflammatory immune responses [30].

In the present study, we have shown a better rate of disease-free survival for patients with TSSCC than for those with BOTSCC. This result is in accordance with results obtained by other authors, which directly compared survival in these two localizations. An analysis by Windon et al. [18] revealed a worse-than-expected prognosis for node-negative patients with HPV-positive BOTSCC. Welters et al. [24] have found better OS for patients with HPV16-positive OPSCC and with the presence of type I specific T-cells, who, as we explained above, probably have TSSCC. These results can be partly explained by a more effective immune system in TSCC than in BOTSCC. However, considering the reasons for a better prognosis of patients with TSSCC than with BOTSCC, it is worth paying attention to the differences between these two localizations in clinical features. In the present study, we have found a higher number of female patients with lower T stages (T2 + 3) among TSSCC compared to BOTSCC. However, we did not notice a difference in the distribution of patients with specific N stages between TSSCC and BOTSCC. In turn, one patient with BOTSCC and HPV16 infection, a woman, 60 years old, with Karnofsky scale < 80%, T3, N2 stages and grade G2, addicted to alcohol and smoking, treated by surgery and adjuvant RT, developed distant metastases 15 months after completing the treatment. These results are partly consistent with those of other authors. Welters et al. [24] revealed lower T and N stages in the cases of patients with cancer type I specific T-cells. In turn, Windon et al. [18], who analyzed 13,081 HPV-positive BOT and 16,874 HPV-positive TS cancers, have shown that patients with HPV-positive BOTSCC were also more frequently male and white and tended to have more advanced TN stages as compared to patients with TSSCC. Contrary to us, these authors did not notice significant differences between these two localizations in the percentage of HPV positivity.

An important limitation of the current study is the small number of patients included in the analysis, as discussed earlier. However, we believe that the presented results present an important signal to study the differences in biology and clinical features between squamous cell carcinomas of the base of the tongue and the tonsils in a greater number of patients. Experimental studies are needed in order to explain the mechanisms concerning the potential differences between these two localizations. Experimental studies are needed in order to explain mechanisms concerning the potential differences between these two localizations. Confirmation of our results in a larger group of patients as well as an indication of biological mechanisms of obtained results, may contribute to the individualization of treatment in OPSCC patients and the use of de-escalation of treatment only in the HPV-positive OSCC group with good prognosis.

## 5. Conclusions

To conclude, due to the relatively small number of 63 patients with SCC of the oropharynx, the present study should be considered a pilot study. However, obtained differences in the biological (in the frequency of HPV infection and the level of viral load) and clinical features between squamous cell carcinomas of the base of the tongue and tonsil may be an important indication for further studies concerning the development of new prognostic and/or predictive factors for patients with HPV16-positive cancers of the oropharynx.

## Figures and Tables

**Figure 1 jpm-13-00361-f001:**
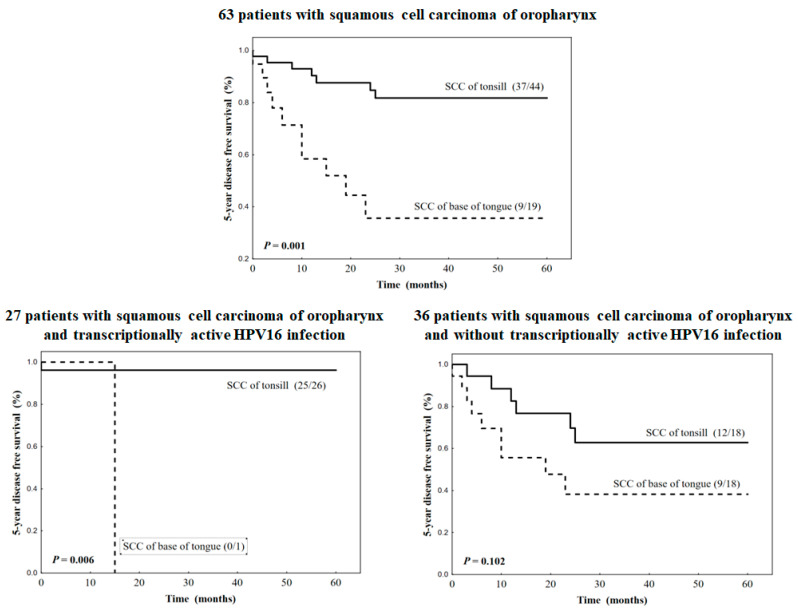
Kaplan–Meier curves concerning disease-free survival stratified by localization of tumors in the group of 63 patients with squamous cell carcinomas of the oropharynx and into two subgroups identified by the prevalence of transitionally HPV16 infection.

**Table 1 jpm-13-00361-t001:** Epidemiological, clinical, and histopathological features in the subgroups of patients with squamous cell carcinoma of base of tongue and tonsil.

	All (%) ^1^	Localization of Squamous Cell Carcinoma		
		Base of Tongue n (%) ^2^	Tonsil n (%)	Test *χ* ^2^ *p* *
All	63 (100.0)	19 (30.2)	44 (69.8)	
Age				
≤58 years ^3^	36 (57.1)	12 (33.3)	24 (66.7)	
>58 years	27 (42.9)	7 (25.9)	20 (74.1)	0.526
Gender				
Female	15 (23.8)	1 (6.7)	14 (93.3)	
Male	48 (76.2)	18 (37.5)	30 (62.5)	**0.023**
Status in the Karnofsky scale				
<80%	26 (41.3)	6 (23.1)	20 (76.9)	
≥80%	37 (58.7)	13 (35.1)	24 (64.9)	0.304
The level of smoking-Brinkman index ^4^				
≤520 ^3^	34 (54.0)	8 (23.5)	26 (76.5)	
>520	29 (46.0)	11 (37.9)	18 (62.1)	0.214
The level of drinking ^5^				
Low	28 (44.4)	5 (17.9)	23 (82.1)	
High	35 (55.6)	14 (40.0)	21 (60.0)	0.057
T stage				
2	15 (23.8)	4 (26.7)	11 (73.3)	
3	32 (50.8)	7 (21.9)	25 (78.1)	
4	16 (25.4)	8 (50.0)	8 (50.0)	**0.045**
N stage				
0	10 (15.9)	5 (50.0)	5 (50.0)	
1	13 (20.6)	2 (15.4)	11 (84.6)	
2 + 3	40 (63.5)	12 (30.0)	28 (70.0)	0.971
Grade				
1	25 (39.7)	7 (28.0)	18 (72.0)	
2 + 3	38 (60.3)	12 (31.6)	26 (68.4)	0.762
Keratinization				
Yes	35 (44.4)	16 (45.7)	19 (54.3)	
No	28 (55.6)	3 (10.7)	25 (89.3)	**0.003**
Transcriptionally active HPV16 infection				
Yes	27 (42.9)	1 (3.7)	26 (96.3)	
Not	36 (57.1)	18 (50.0)	18 (50.0)	**0.000**
P16 immunopositivity				
Yes	27 (42.9)	7 (25.9)	20 (74.1)	
Not	36 (57.1)	20 (55.6)	16 (44.4)	**0.019**
Viral load (only in HPV16 positive, n = 27)				
>6764.3 copies/cell ^3^	17 (63.0)	0 (0.0)	17 (100.0)	
≤6764.3 copies/cell	10 (37.0)	4 (40.0)	6 (60.0)	**0.024 ***
Physical genome status (only in HPV16 positive, n = 27)				
Integrated	8 (29.6)	0 (0.0)	8 (100.0)	
Mixed	19 (70.4)	4 (21.1)	15 (78.9)	0.508
Treatment				
Definitive CisPt-CRT or surgery + CisPt-CRT	28 (44.4)	4 (14.3)	24 (85.7)	
Definitive RT or surgery + RT	19 (30.2)	8 (42.1)	11 (57.9)	
Induction CT + definitive RT	16 (25.4)	7 (43.7)	9 (56.3)	0.058
Treatment outcome				
Regression of cancer disease	45 (71.4)	9 (20.0)	36 (80.0)	
Treatment failure	2 (3.2)	1 (50.0)	1 (50.0)	
Local recurrence	12 (19.1)	7 (58.3)	5 (41.7)	
Distant metastases	2 (6.3)	1 (50.0)	1 (50.0)	**0.049**
Survival				
Alive at the last follow-up	34 (54.0)	3 (8.8)	31 (91.2)	
Death from cancer disease	15 (20.0)	9 (60.0)	6 (40.0)	
Death from other reasons	14 (35.7)	7 (50.0)	7 (50.0)	**0.000**

Abbreviations: CisPt-CRT, concurrent chemoradiotherapy with cisplatin; CT, chemotherapy; ^1^ Column percentage; ^2^ Row percentage; ^3^ Median value; ^4^ Number of cigarettes per day × years of smoking; ^5^ Low level of drinking—no alcohol and occasional drinkers (at most two drinks a day, especially with a meal) high level of drinking—more than 15 drinks high percentage alcohol in a week and alcoholics; * The statistical significance limit for *p* value was accepted as *p* < 0.05, significant results are written in bold font.

**Table 2 jpm-13-00361-t002:** 5-year disease-free survival in the group of 63 patients with squamous cell carcinoma of the head and neck and subgroup of 27 patients with transcriptionally active HPV16 infection according to tumor localization.

	5-Year Disease Free Survival			
	Response n (%)	HR	95% CI	Log-Rank *p*
63 patients with squamous cell carcinoma of head and neck				
Base of tongue	9/19 (47.4)	5.138	1.927–13.706	0.001
Tonsil	37/44 (84.1)	1.000		
27 patients with squamous cell carcinoma of head and neck and transcriptionally active HPV16 infection				
Base of tongue	0/1 (0.0)	12.220	1.378–25.283	0.006
Tonsil	25/26 (96.1)	1.000		
36 patients with squamous cell carcinoma of head and neck and without transcriptionally active HPV16 infection				
Base of tongue	9/18 (50.0%)	2.372	0.836–6.732	0.102
Tonsil	12/18 (66.7)	1.000		

## Data Availability

All data generated or analyzed during this study are included in this article. Further enquiries can be directed to the corresponding author.

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
