# Peer review of "Differences between Squamous Cell Carcinomas of the Base of the Tongue and the Tonsils in Prevalence of HPV16 Infection, Its Type, and Clinical Features"

_jpm, 2023, doi:10.3390/jpm13020361_

Round 1
Reviewer 1 Report
thank you for inviting me to review this manuscript on the differences in HPV infection between sites in the oropharynx.
Introduction
- I cannot see a [1] reference; the first I can see is [2] at line 39.
- 'during last years' detection of HPV - this phrasing makes it sound as though this is quite recent, whereas as shown in the reference you've selected it has been known for almost 20 years. I think the phrasing should be reworded to reflect this.
- in line 53 you reference [3] please confirm this is correct, I do not see anywhere in that article reference to 'more than 40%', It is a review article.
- adding a section as the goals of deescalation would be of value
- include what you predict active HPV infection to have on prognosis
Method
- there does not appear to be any ethics statement re approval board in the methods, only at the conclusion of the paper.
- please clarify why the inclusion criteria states that participants had a follow up minimum 5 years, but the mean follow up page 5 states there was a mean of 42.0
Results
- page 7 correct spelling of localisation. currently it is spelt 'lacalisation'
Discussion
- the first sentence does not make sense. please rephrase
- spelling error page 7, line 240 - should be 'noticed' not 'notoced'
- the sentences are all very long through the discussion and challenging to follow. I would suggest re-phrasing to make it clearer to follow.
- do not overuse phrases like 'in turn'. it becomes repetitive. eg in paragraph 2 of page 8 it is used 3 times.
- syntax and grammar need work through this section eg page 7 line 240 'what suggest better' - revise. also line 258 of page 8, page 8 line 301 - should be 'explained'
- page 7 line 246 'other authors' please reference what studies yours aggrees with. this should also be referenced when you refer to this concept again page 8 line 297
- you mention the number of patients being a limitation, however this is not linked to any concepts through the rest of the paper, it appears a nominal mention at the end. I would suggest these findings are exploratory, and the significant limitation imposed by the low number is discussed earlier. It is also retrospective, and essentially a secondary analysis of an already examined patient group.
Author Response
Reviewer #1
We highly appreciate the Reviewer’ insightful and helpful comments on our manuscript.
Introduction
- I cannot see a [1] reference; the first I can see is [2] at line 39.
You have right. The [1] reference is not necessary, therefore in revised version of manuscript we removed this publication from the list of references and consequently renumbered the references throughout the whole manuscript
-during last years' detection of HPV - this phrasing makes it sound as though this is quite recent, whereas as shown in the reference you've selected it has been known for almost 20 years. I think the phrasing should be reworded to reflect this.
According to the Reviewer suggestion we reworded this pharse as follows “during the last few decades, it was noticed increase in incidence rates for SCC of tonsil (TSSCC) and base of tongue (BOTSCC), what probably is related with increased number of HPV infection in these anatomical sites [1].”
We also changed the references here to a more up-to-date one, that is:
Jordan KH, Fisher JL, Paskett ED. Distinct sociodemographic differences in incidence and survival rates for human papillomavirus (HPV)-like, non-HPV-like, and "other"-like oral cavity and pharynx cancers: An analysis of Surveillance, Epidemiology and End Results (SEER) Program data. Front Oncol. 2022 Aug 18;12:980900. doi: 10.3389/fonc.2022.980900.
- in line 53 you reference [3] please confirm this is correct, I do not see anywhere in that article reference to 'more than 40%', It is a review article.
According to the Reviewer suggestion we cited in this place results of 4 meta-analysis, which are summarized in the table below and from which it is clear that in more than 40% of patients with HPV16 positive OPSCC progression of cancer disease.
- Dayyani F., Etzel C.J., Liu M., Ho C.H., Lippman S.M., Tsao A.S. Meta-analysis of the impact of human papillomavirus (HPV) on cancer risk and overall survival in head and neck squamous cell carcinomas (HNSCC) Head Neck Oncol. 2010;29:2–15. doi: 10.1186/1758-3284-2-15.
- Liu H., Li J., Zhou Y., Hu Q., Zeng Y., Mohammadreza M.M. Human papillomavirus as a favorable prognostic factor in a subset of head and neck squamous cell carcinomas: a meta-analysis. J. Med. Virol. 2016;89(4):710–725. doi: 10.1002/jmv.24670.
- O’Rorke M.A., Ellison M.V., Murray L.J., Moran M., James J., Anderson L.A. Human papillomavirus related head and neck cancer survival: a systematic review and meta-analysis. Oral Oncol. 2012;48(12):1191–1201. doi: 10.1016/j.oraloncology.2012.06.019.
- Ragin C.C., Taioli E. Survival of squamous cell carcinoma of the head and neck in relation to human papillomavirus infection: review and meta-analysis. Int. J. Cancer. 2007;121(8):1813–1820. doi: 10.1002/ijc.22851.
Because we included these four papers in the References, we consequently change numbers of others references.
- adding a section as the goals of deescalation would be of value
According to the Reviewer's suggestion, we briefly indicated that it is to avoid toxic reactions after cisplatin or/ and radiotherapy treatment. (page 2, line 3)
- include what you predict active HPV infection to have on prognosis
According to the Reviewer's suggestion, we added the following sentences in tha last paragraph of Introduction:
“It should be noticed that detection of HPV infection based only on the analysis of virus DNA or P16 expression may lead to obtain false positive results. HPV DNA presence does not indicate whether HPV is transcriptionally active or not. P16, in turn, is a surrogate marker indicating active HPV infection, but its overexpression may not exactly match the HPV DNA, because it may be also caused by other, non-viral factors.”
Method
- there does not appear to be any ethics statement re approval board in the methods, only at the conclusion of the paper.
This is in accordance with the Instructions for Authors OF JPM (https://www.mdpi.com/journal/jpm/instructions).
- please clarify why the inclusion criteria states that participants had a follow up minimum 5 years, but the mean follow up page 5 states there was a mean of 42.0
The mean follow-up time was 42.0 months, because in some patients cancer progression (treatment failure, locoregional recurrence, distant metastasis) was noticed less than 5 years after treatment completing.
Results
- page 7 correct spelling of localisation. currently it is spelt 'lacalisation'
Corrected as indicated by the Reviewer.
Discussion
- the first sentence does not make sense. please rephrase
Corrected as suggested by the Reviewer as follows:.
To the best of our knowledge, this is the first report showing the difference in the level of viral load between SCC of base of tongue and tonsil.
- spelling error page 7, line 240 - should be 'noticed' not 'notoced'
Corrected as suggested by the Reviewer.
- the sentences are all very long through the discussion and challenging to follow. I would suggest re-phrasing to make it clearer to follow.
According to the Reviewer suggestion many sentences of the discussion have been carefully rewritten or reorganized to enhance the logic flow.
- do not overuse phrases like 'in turn'. it becomes repetitive. eg in paragraph 2 of page 8 it is used 3 times.
According to the Reviewer suggestion, we refrained from repeating “in turn”, in this paragraph.
- syntax and grammar need work through this section eg page 7 line 240 'what suggest better' - revise. also line 258 of page 8, page 8 line 301 - should be 'explained'
We hope that we corrected all syntax and grammar errors.
- page 7 line 246 'other authors' please reference what studies yours aggrees with. this should also be referenced when you refer to this concept again page 8 line 297
According to the Reviewer suggestion, we added the references in above-mentioned places.
- you mention the number of patients being a limitation, however this is not linked to any concepts through the rest of the paper, it appears a nominal mention at the end. I would suggest these findings are exploratory, and the significant limitation imposed by the low number is discussed earlier. It is also retrospective, and essentially a secondary analysis of an already examined patient group.
According to the Reviewer suggestion, the last paragraph of discussion was rewriting as follows:
“Important limitation of the current study is small number of patients included into the analysis, what was discussed earlier. However, we believe that presented results are important signal to study the differences in biology and clinical features between squamous cell carcinomas of base of tongue and tonsil on more number of patients. Experimental studies are needed in order to explain mechanisms concerning the potential differences between these two localizations. Confirmation of our results in a larger group of patients as well as indication of biological mechanisms of obtained results may contribute to the individualization of treatment in OPSCC patients and the use of de-escalation of treatment only in the HPV-positive OSCC group with good prognosis.”

Reviewer 2 Report
Dear authors,
Thanks for your exceptional effort
A few remarks to be noted
1- Title: needs to be re-structured. Too lengthy and dissociating
2- Methodology:
- The institute's protocol should be explained. What were the criteria for choosing surgery for a patient and abandoning it for the other?
- Having only one patient with BOTSCC who had transcriptionally active HPV16 infection can never be a basis for any conclusion apart from the prevalence
- More details should be added regarding that patient
3- Tables:
Table 1:
- Why was 58 years chosen as a limit for age groups?. I suggest separation into 20-40, 40-60, >60
- N0 & N1 can not be considered the same. Please separate
- The same for T2 & T3
Discussion:
should focus more on the epidemiological & prognostic factors in addition to the difference in prevalence between the two sites. But as mentioned before, no true conclusions can be built on only one patient
Author Response
We are very grateful to you for your insightful comments.
Dear authors,
Thanks for your exceptional effort
A few remarks to be noted
- Title: needs to be re-structured. Too lengthy and dissociating
According to the Reviewer's suggestion, we changed the title of the paper, as follows:
“Differences between squamous cell carcinomas of the base of tongue and tonsil in prevalence of HPV16 infection, its type and clinical features”
2- Methodology:
- The institute's protocol should be explained. What were the criteria for choosing surgery for a patient and abandoning it for the other?
The decision regarding the operation was made by a team of physicians consisting of a surgeon, radiotherapist and chemotherapist, who referred for the surgery patients with advanced disease (T≥3) and histological grade G1. This information is added in the paragraph 3.1. Clinical characteristics of a group of 63 patients with squamous cell carcinoma of the oropharynx of the Results.
- Having only one patient with BOTSCC who had transcriptionally active HPV16 infection can never be a basis for any conclusion apart from the prevalence
We agree with the Reviewer, we were surprised ourselves that only one patients with base of tongue had HPV16 infection (particularly in the light of results of meta-analysis of Nadiye et al.). However, in the Discussion paragraph, we wrote that this results should be treated with caution and require confirmation in a larger group of patients. Besides, concluding we underlie that our results may be a impulse to perform experimental studies in order to confirm or explain our observation. So, we did not conclude definitely that HPV16 active infection is more common in TSSCC, therefore we believe that the presented results indicate further research directions.
- More details should be added regarding that patient
According to the Reviewer suggestion we added the following sentence:
“One patient with BOTSCC and HPV16 infection, it is a woman, 60 years old, with Karnofsky scale < 80%, T3, N2 stages and grade G2, addicted to alcohol and smoking, treated by surgery and adjuvant RT, who developed distant metastases 15 months after completing the treatment.”
in the paragraph 3.2. Differences between squamous cell carcinoma of base of tongue and tonsil in epidemiological, clinical and histopathological features as well as prevalence of HPV16 infection and its type of the Results.
3- Tables:
Table 1:
- Why was 58 years chosen as a limit for age groups?. I suggest separation into 20-40, 40-60, >60
In the description of the table, we explain that this is the median age in the study group. We did not decide to separate this group according to ranges proposed by the Reviewer because of low numbers of patients into the specific range.
- N0 & N1 can not be considered the same. Please separate
- The same for T2 & T3
According to the Reviewer suggestion, in the Table 1 we separate N0 and N1 or T2 and T3
Discussion:
should focus more on the epidemiological & prognostic factors in addition to the difference in prevalence between the two sites. But as mentioned before, no true conclusions can be built on only one patient.
We added information about epidemiological and prognostic features of one patients with BOTSCC and active HPV16 infection in last paragraph of Discussion. In turn, in the Conclusion we underlie that our study should be treated as a pilot study.

Reviewer 3 Report
This study has serious methodological flaws. The authors use commercial kits to demonstrate the presence of oncogenic HPV DNA types in formalin fixed biopsies but use immunochemistry for p16 to confirm HPV gene expression - this is their definition of transcriptionally active. This is not an unequivocal demonstration of HPV 16 E7 gene expression but a useful laboratory test, helpful un a clinical setting but not acceptable for serious studies. RT PCR for early HPV 16 gene expression would be more convincing. The key question for viral gene expression in these cancers is episomal versus integrated species, viral gene expression may be being driven entirely by an integrant even in the presence of multiple episomal copies. Data on viral load in the absence of evidence for integration is not informative.
Author Response
This study has serious methodological flaws. The authors use commercial kits to demonstrate the presence of oncogenic HPV DNA types in formalin fixed biopsies but use immunochemistry for p16 to confirm HPV gene expression - this is their definition of transcriptionally active. This is not an unequivocal demonstration of HPV 16 E7 gene expression but a useful laboratory test, helpful un a clinical setting but not acceptable for serious studies. RT PCR for early HPV 16 gene expression would be more convincing. The key question for viral gene expression in these cancers is episomal versus integrated species, viral gene expression may be being driven entirely by an integrant even in the presence of multiple episomal copies. Data on viral load in the absence of evidence for integration is not informative.
In regard to the possibility of analysis of E6/E7 expression by immunohistochemistry (IHC) we would like to emphasize that it is not possible in our group of patients, because FFPE from our patient’s group were used in three projects (mentioned in the paper). Therefore, it is not possible to cut them further, due to the necessity of tissue preservation in the FFPE. We realize that the identification of the transcripts of the viral oncogenes E6/E7, implicated in the oncogenic process, through mRNA techniques is assumed as gold-standard test to elucidate the oncogenic role of HPV in the tumour. However, in case of archival paraffin blocs it may be hampered by difficulties in isolating RNA of sufficient quality. Other possibility is to analyse immunohistochemical expression of HPV E6 E7 proteins. However, protocol of immunohistochemical staining of E6 and EE7 is not well established. There is a few of E6 and E7 antibodies. As indicate from the paper of Stiansny et al. [2016] using different antibodies (Abcam, Chemicon, Invitogen, Biorybt) there are significant differences concerning E6 and E7 immunoreactivity in cervical cancers. In turn, in the study of Islam et al. [2021] four novel variations in the HPV16 E6 region and two novel E6^E7*I and E6^E7*II fusion transcript variants were identified. On the basis of in silico analysis these authors underlie that there are significant differences in stability of novel E6 variants, what means that some part of results of IHC analysis of E6/E7 expression may be false negative. Therefore, to analyse transcriptionally active HPV infection we decided to combined two techniques: P16 expression and analysis of HPVDNA presence. Such combination is considered as surrogate marker of transcriptionally active viral infection.
We decided to use the term “active HPV16 infection” in concordance with other authors, i. e. Ndiaye C, Mena M, Alemany L, Arbyn M, Castellsagué X, Laporte L et al (2014) HPV DNA, E6/E7 mRNA, and p16INK4a detection in head and neck cancers: a systematic review and meta-analysis. Lancet Oncol 15(12):1319–1331 and Rietbergen MM, Leemans CR, Bloemena E, Heideman DA, Braakhuis BJ, Hesselink AT, Witte BI, Baatenburg de Jong RJ, Meijer CJ, Snijders PJ, Brakenhoff RH. Increasing prevalence rates of HPV attributable oropharyngeal squamous cell carcinomas in the Netherlands as assessed by a validated test algorithm. Int J Cancer. 2013 Apr 1;132(7):1565-71. doi: 10.1002/ijc.27821, who underlie that “the use of inmunohistochemical techniques to detect p16 INK4a followed by HPV DNA detection has been also validated to clinically detect an oncogenically active HPV infection in oropharyngeal cancer” and that “The availability of a reliable test algorithm for FFPE tumor specimen is needed to compare prevalence rates of HPV-positive OPSCCs around the world. In addition, a reliable HPV detection method, that allows evaluation of active, oncogenic HPV involvement in OPSCC, is of major importance for the selection of patients in clinical de-escalation trials”.
We have data on HPV16 genome status and present them in Table 1 of our paper. As you can see all BOTSCC were characterized by lower level of viral load and simultaneously mixed status of viral genome.

Reviewer 4 Report
Manuscript Number: jpm-2185366-peer-review-v1
Title: Differences between squamous cell carcinomas of the base of 2 tongue and tonsil in prevalence of HPV16 infection and its type 3 as well as in epidemiological, clinical and histopathological 4 features
1. Yes, this subject is useful for publication in JPM.
2. Authors compared the incidence of transcriptionally active HPV16 infection and its type as well as other epidemiological, clinical and histopathological features between SCC of base of tongue and tonsil.
3. The design, methods, and results are clearly presented.
4. Discussion is logical and correct. Line 243 two times should be
5. Conclusion is too short.
6. References are current and pertinent.
This paper should be published after minor correction. Correction of English and conclusion.
.
Author Response
Manuscript Number: jpm-2185366-peer-review-v1
Title: Differences between squamous cell carcinomas of the base of 2 tongue and tonsil in prevalence of HPV16 infection and its type 3 as well as in epidemiological, clinical and histopathological 4 features
- Yes, this subject is useful for publication in JPM.
Authors compared the incidence of transcriptionally active HPV16 infection and its type as well as other epidemiological, clinical and histopathological features between SCC of base of tongue and tonsil.- The design, methods, and results are clearly presented.
- Discussion is logical and correct. Line 243 two times should be
Thank you. We have corrected this error
- Conclusion is too short.
„However, due to the relative small number of 63 patients with SCC of oropharynx, the present study should be considered as a pilot study. Therefore, the further clinical and in vitro or preclinical studies are necessarily to explain role of CD44 and Sox-2 in mechanism of HPV positive and HPV negative response to cancer therapy.”
- References are current and pertinent.
This paper should be published after minor correction. Correction of English and conclusion.

Reviewer 5 Report
Though there is enough evidence available in literature about the difference in outcome between HPV positive and negative oropharyngeal cancer, the novelty of this study lies in identifying the subset of better performing HPV positive tumours.
However, results need to be depicted in a better way. The authors can provide kaplan Meier analysis of tonsil and base of tongue cancers stratified by HPV status and viral load.
Author Response
Thank you for evaluating our work
Though there is enough evidence available in literature about the difference in outcome between HPV positive and negative oropharyngeal cancer, the novelty of this study lies in identifying the subset of better performing HPV positive tumours.
However, results need to be depicted in a better way. The authors can provide kaplan Meier analysis of tonsil and base of tongue cancers stratified by HPV status and viral load.
According to the Reviewer suggestion we added KM plots as Fig. 1.

Round 2
Reviewer 1 Report
This current manuscript version represents a marked improvement I have no further additions
Reviewer 3 Report
The authors have addressed my concerns